# Triboemission of FINE and Ultrafine Aerosol Particles: A New Approach for Measurement and Accurate Quantification

**Roman Nevshupa [1],\*** , **Marta Castellote [1]** , **Jesus Antonio Carlos Cornelio [2]**
**and Alejandro Toro [2]**

[1] Spanish National Research Council (IETCC-CSIC), C/Serrano Galvache 4, Madrid 28033, Spain; marta.castellote@csic.es

[2] Tribology and Surfaces Group, Universidad Nacional de Colombia, Medellín 055040, Colombia; iqcornelio@gmail.com (J.A.C.C.); aotoro@unal.edu.co (A.T.)

\* Correspondence: r.nevshupa@csic.es; Tel.: +34-913-020-440

**Abstract:** A dynamic model based on mass balance of fine aerosol particles was developed in order to tackle the problem of accurate quantification of mechanically stimulated particle emission (MSPE) from nanofunctionalized and solid lubricating materials. In contrast to the conventional approach, the model accounts for the effect of air turbulization caused by moving parts of the experimental tribological setup on the enhancement of particle deposition velocity. The increase of the velocity of the moving parts results in an increase of the deposition velocity that leads to a significant underestimation of experimentally measured particle emission rates. The developed model was experimentally verified using natural and artificial nanoparticle aerosols. Finally, the new methodology of particle emission rate quantification was employed for the analysis of fine particle emission produced when the solid lubricating materials were tested against a sliding steel surface. The developed method paves the way for defining a standard method of experimental assessment of nanoparticle triboemission enabling the experimental results obtained in various laboratories to be compared. It also bridges the gap between the phenomenological models and experimental measurements.

**Keywords:** triboemission; abrasion; solid lubricants; fine particle aerosol

---

## 1. Introduction

Since the beginning of the broad introduction of engineered nanomaterials (ENMs) in various applications and consumer products, there has been a concern over the potential risks of release of these materials into the environment [1–3]. For quite a long time, mechanical solicitations such as machining [4], erosion, abrasion, sanding, rubbing (including brake, road and tyre wear) and weathering have been considered the most typical liberation processes responsible for non-exhaust particle emission [5–7], although the reported experimental results are surrounded by controversy [8,9]. Friction can be responsible for significant emissions of fine aerosol particles even from conventional materials, which do not have embedded nanoparticles, e.g., friction stir welding [10], friction between railway brake disks and pads [11].

Experimental measurement of time series of particle emissions from various materials subjected to mechanical solicitation is of high interest for quantitative evaluation of the emission characteristics [12]. The early studies employed the experimental setups being a simple combination of the aerosol generator and an aerosol measurement system. The aerosol generators were a standard Taber abrader or a sander, whereas the aerosol measurement system normally consisted of a simple non-airtight hood surrounding the mechanically affected zone [8,10,13,14] and a standard aerosol measurement

apparatus. The number concentration and number size distribution of aerosols were usually reported. However, the obtained data are device specific and cannot be used for comparison of the emission characteristics among various studies. In a noteworthy review [6], Koivisto et al. highlighted that the particle emission rates reported in literature varied over six orders of magnitude, and they attributed such dispersion to the absence of harmonized experimental methodology, differences in sampling efficiency and insufficient air mixing in the experimental setup. This problem was addressed by various groups and it was partly solved when a tight (or nearly tight) aerosol chamber [15–17] was set to isolate the local environment around the tribological contact from the undesired particle dispersion in the ambient air and the influence of motors and drives on particle generation/deposition [18]. A similar configuration, but on a significantly larger scale, was applied for measuring aerosols generated at simulated tire–road contacts [19,20]. Measurement accuracy is of high importance since the data are used to assess particle exposure scenarios and the health impacts through appropriate models of particle dynamics based on the analysis of the relevant processes such as particle exchange with the environment, coagulation, deposition through Brownian and turbulent diffusion, gravitation settling and turbophoresis [21,22]. One of the possible reasons for the large dispersion of the reported results can be related to inappropriate design of the experimental setup or inadequate experimental procedures, which do not account for air turbulization by the moving parts. Although the motors and drives situate outside the aerosol chamber, the rotating tool, reciprocating abrader, etc., may cause increasing rates of particle deposition. Our preliminary experiment provides clear evidence for this phenomenon. In this experiment, the aerosol concentration in air was measured during rotation of the sample with and without rubbing (Figure 1). When the sample was rubbed by a steel bar, "positive" aerosol emission, i.e., increase of airborne nanoparticle concentration, was observed. However, when the sample was rotated without rubbing, the concentration of airborne particles decreased. This "negative" emission was due to enhanced deposition of residual airborne particles in the aerosol chamber. In filtered, nearly particle-free air [17], the effect is obscure, which may explain why it has remained inconspicuous for researchers for a long time.

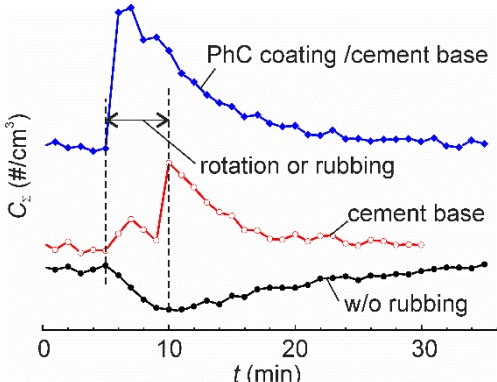

**Figure 1.** The time series of total airborne particle concentration in the aerosol chamber of a pin-on-disk tribometer for three tests: rotation of the turnable table without rubbing, rubbing of cement base sample, and rubbing of a nanofunctionalized photocatalytic (PhC) coating on a cement base. In the latter two tests, the samples were rubbed by a steel bar 10 mm in diameter, normal load 13.2 N. Rotation velocity was 600 rpm that corresponded to 1 m/s linear sliding speed. Vertical dashed lines show the period of rotation or rubbing. All the tests were carried out in normal atmosphere with the total ambient particle concentration 3000–4000 cm$^{-3}$ in the particle size range 11–365 nm.

Quantification of the particle emission for various materials and experimental conditions is also the first step towards achieving systematic knowledge of this complex phenomenon, which is not well understood yet. For this purpose, an advanced model of concentration dynamics of airborne particles should be developed. So far, the Eulerian approach has been the basis of the modelling of dynamics of particles dispersed throughout a fluid and their deposition [23]. Although it was successfully applied

to calculations of the particle deposition onto the ducts [21,24,25], small rooms [26], static manikin [27] and other objects of relatively simple geometry, the Eulerian approach is less suitable for tribological applications, which are characterized by a much more complex geometry of the experimental setup, relative motion of the components, and complex and transient turbulent flows. A less accurate, but much simpler and more practical phenomenological model [26] can be a good alternative to the Eulerian model.

In this work, we developed a new phenomenological model based on the equation of a mass balance of aerosol particles in an aerosol chamber. To account for the accelerated particle deposition caused by moving components of the setup, two new empirical parameters were introduced. They depend upon the particle size, geometry and velocity of the moving parts. A practically significant method for assessment of these parameters has been also developed. This method is based on the measurement of time constants of the transients of particle concentrations under different experimental conditions. The model was verified using natural and artificial fine particle aerosols. Finally, the developed model was used to determine the real particle emission rate and the number of emitted particles in a case study of rubbing of a nanofunctionalized solid lubricating bar of friction modifier against a steel disk.

## 2. Model

A schematic drawing of a typical experimental setup for characterization of particle emission from ENMs subjected to abrasion under rotation is shown in Figure 2. An aerosol chamber is connected to the baseplate using a chamber seal. A rotatable table is placed inside the aerosol chamber and secured to the motor shaft. The shaft is sealed at its pass through the hole in the baseplate to avoid air and particle exchange between the chamber and the exterior. The chamber has small openings to allow the surrounding air from the controlled environment to enter the chamber and to compensate the pressure drop due to air sampling by the particle counter. Because of labyrinth geometry, these openings impede the particles from leaving the chamber, but the ambient airborne particles can enter with the inlet flow. A sample of material to be studied is fixed to the table and can be subjected to mechanical action by the tool or indenter. The chamber is also equipped with a port for connection of a particle counter. In this work we did not aim figuring out the physical mechanisms by which the aerosol particles are deposited on the surfaces of experimental setup facing the aerosol chamber. The particle deposition was looked at as an unidentified process resulting in the decrease of particle concentration in the air. Likely, the particle emission was considered as an increase of the particle concentration in the air irrespectively of how the particles were emitted.

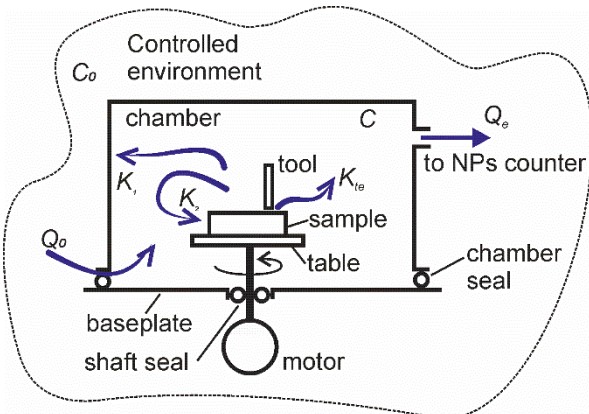

**Figure 2.** Schematic drawing of the experimental setup and particle flows.

The setup is situated in an environment chamber, in which the concentration of ambient airborne particles $C_{0,i}$ can be controlled by air recirculation through a filtration system. In some tests, air filtration was off, and unfiltered air was used to study the deposition velocity of ambient particles. The following

particle flows are considered in the model: $K_{1,i}$ is the rate of particle deposition in the setup at a standstill, $K_{2,i}$ is the rate of particle deposition in the setup in motion, $K_{te,i}$ is the rate of mechanically stimulated particle emission (MSPE) from the material subjected to abrasion or rubbing. The subscript *i* stands for a fraction of the corresponding parameter at the particles size-section *i*. The emitted particles can take their origin from wearing of one or both of the contacting surfaces. Likely they can be formed in tribo- and photochemical processes associated with frictional activation of materials. The knowledge of chemical and physical properties of the emitted aerosol particles is not required for the accurate quantification of the emission rates, but it can be helpful for reducing the required experimental efforts.

For the number concentrations of airborne particles below $10^4$ cm$^{-3}$, particle coagulation or agglomeration effects can reasonably be neglected [26]. The particle balance in the aerosol chamber for two specific cases of the setup at a standstill and the setup in motion as well as the general model are described below.

## 2.1. Setup at a Standstill

With the movable parts at a standstill, the equation of balance of aerosol particles can be written in the following form:

$$\frac{dN_i}{dt} = K_{0,i} - K_{NP,i} - K_{1,i}, \tag{1}$$

where $N_i(t)$ is the total number of particles in the aerosol chamber, $K_{0,i}$ is the rate of particles entrance into the chamber from the environment, $K_{NP,i}$ is the rate of particles leave from the chamber to the particles counter.

For fully mixed aerosols:

$$N_i(t) = C_i(t)V, \tag{2}$$

where $V$ is the volume of the aerosol chamber and $C_i$ is the number concentration of particles at size-section *i*.

The terms on the right side of (1) are proportional to the particles concentrations in the chamber $C_i$ or in the environment $C_{0,i}$:

$$K_{0,i} = C_{0,i}Q_0, \tag{3}$$

$$K_{NP,i} = C_i Q_{NP}, \tag{4}$$

where $Q_0$ is the air flow into the chamber through the openings, $Q_{NP}$ is the air flow from the chamber to the particles counter. $K_{1,i}$ accounts for particle eddy diffusion in the boundary layer at the surfaces, turbophoretic deposition and settling [25] and it can be found by integration of the total local particle flux toward the surface (along axis *y*) over whole internal surface area, *S*, of the experimental setup:

$$K_{1,i} = \int_S \left( V_t C_i(t) + iV_s C_i(t) - (\varepsilon_p + D)\frac{\partial C_i}{\partial y} \right) dS, \tag{5}$$

where $V_t$ is the turbophoretic velocity, $V_s$ is the settling velocity, *i* is used to characterize the orientation of the surface, $\varepsilon_p$ is the particle eddy diffusivity in the boundary layer, $D$ is the Brownian diffusivity of the particle. All terms under the integration sign are proportional to $C_i(t)$, which, for well mixed aerosols, is spatially independent. Thus, $C_i(t)$ can be pulled out from the integral:

$$K_{1,i} = C_i(t) \int_S \left( V_t + iV_s - (\varepsilon_p + D)\frac{1}{C_i}\frac{\partial C_i}{\partial y} \right) dS \tag{6}$$

Now, the integral in (6) depends on the distribution of air and particle velocities, diffusion constants of the particles and normalized gradient of particle concentration in the boundary layer, which, in turn, is a function of the setup geometry, particle properties and distribution of air flows.

For the given setup and experimental conditions all these parameters are nearly constant, and the expression (6) can be rewritten:

$$K_{1,i} = C_i(t)A_{1,i}, \tag{7}$$

where $A_{1,i}$ is the empirical particle deposition velocity on the internal surfaces of the setup at a standstill.

Expression (1) can be transformed to the following ordinary linear differential equation:

$$V\frac{dC_i}{dt} = Q_0 C_{0,i} - C_i(t)(Q_0 + A_{1,i}), \tag{8}$$

which solution is a single exponential function:

$$C_i(t) = \frac{C_{0,i}Q_0}{Q_0 + A_{1,i}} + C_2 \exp\left(-\frac{Q_0 + A_{1,i}}{V}t\right), \tag{9}$$

where $C_2$ is the integration constant, which depends on the initial conditions. The time constant of the concentration decay is a function of the air flow pumped by the particle counter, deposition efficiency and the chamber volume:

$$\tau_{1,i} = \frac{V}{Q_0 + A_{1,i}}. \tag{10}$$

Function (9) is similar to the empirical function suggested by Morency et al. [26].

A limit of (9) at $t\to\infty$ gives the equilibrium particle concentration:

$$C_{1,i}^{eq} = C_{0,i}\frac{Q_0}{Q_0 + A_{1,i}} \tag{11}$$

Expressions (10) and (11) can be used to determine the empirical deposition efficiency from the time constant and experimental parameters:

$$A_{1,i} = \frac{V}{\tau_{1,i}} - Q_0, \tag{12}$$

$$A_{1,i} = Q_0\left(\frac{C_{0,i}}{C_{1,i}^{eq}} - 1\right) \tag{13}$$

The corresponding uncertainties for (12) and (13):

$$u(A_{1,i}) = \sqrt{\frac{\Delta V^2}{6\tau_{1,i}^2} + \frac{\Delta Q_0^2}{6} + \frac{V^2}{\tau_{1,i}^4}u^2(\tau_{1,i})}, \tag{14}$$

$$u(A_{1,i}) = \sqrt{\left(\frac{C_{0,i}}{C_{1,i}^{eq}} - 1\right)^2 \frac{\Delta Q_0^2}{6} + \left(\frac{Q_0}{C_{1,i}^{eq}}\right)^2 u^2(C_{0,i}) + \frac{(Q_0 C_{0,i})^2}{\left(C_{1,i}^{eq}\right)^4}u^2\left(C_{1,i}^{eq}\right)}, \tag{15}$$

where $\Delta V$ is the precision of the volume measurement, $\Delta Q_0$ is the precision of the air flow of the particle counter, $u()$ is the measurement uncertainty of the corresponding variable.

## 2.2. Setup in Motion, without MSPE

Motion of the elements inside the aerosol chamber causes changes of the air velocity and, at higher Reynolds numbers, turbulization, which affects the particle deposition velocity:

$$A_{2,i} = \int_S \left(V_t + iV_s - (\varepsilon_p + D)\frac{1}{C_i}\frac{\partial C_i}{\partial y}\right)dS. \tag{16}$$

It is reasonable to assume that the air flow changes only on a portion of the internal surfaces, $S_2$, and it remains unchanged on the rest of the surfaces, $S_1$. $S_1$ and $S_2$ are the non-overlapping subareas of the internal surfaces: $S = S_1 + S_2$. Then, the integration in (13) can be done on $S_1$ and $S_2$ separately yielding two partial particle deposition velocities, $A_{1,i}^{S_1}$ and $A_{2,i}^{S_2}$:

$$A_{2,i} = \int_{S1}\left(V_t + iV_s - (\varepsilon_p + D)\frac{1}{C_i}\frac{\partial C_i}{\partial y}\right)dS + \int_{S2}\left(V_t + iV_s - (\varepsilon_p + D)\frac{1}{C_i}\frac{\partial C_i}{\partial y}\right)dS = A_{1,i}^{S_1} + A_{2,i}^{S_2}. \quad (17)$$

The solution of (17) can be found through finite-element analysis of turbulent flows in complex geometries that is a quite difficult and time-consuming task. For practical purposes, we suggest the following simplified empirical approach. The increment of particle deposition velocity on $S_2$ can be expressed as $\Delta A_{2,i}^{S_2}$, then (17) can be rewritten:

$$A_{2,i} = A_{1,i}^{S_1} + A_{1,i}^{S_2} + \Delta A_{2,i}^{S_2} = A_{1,i} + \Delta A_{2,i}^{S_2}. \quad (18)$$

For the particle deposition velocity in form (18), the solution of (1) is the following exponential function (exponential approach):

$$C_i(t) = \frac{C_{0,i}Q_0}{Q_0 + A_{1,i} + \Delta A_{2,i}^{S_2}} + C_3 \exp\left(-\frac{Q_0 + A_{1,i} + \Delta A_{2,i}^{S_2}}{V}t\right), \quad (19)$$

where $C_3 = C_i(0) - \frac{C_{0,i}Q_0}{Q_0 + A_{1,i} + \Delta A_{2,i}^{S_2}}$ is the integration constant, which depends on the initial condition ($C_i(0)$).

The time constant of particle deposition:

$$\tau_{2,i} = \frac{V}{Q_0 + A_{2,i}} \quad (20)$$

and the equilibrium particle concentration (at $t \to \infty$):

$$C_{2,i}^{eq} = C_{0,i}\frac{Q_0}{Q_0 + A_{1,i} + \Delta A_{2,i}^{S_2}}. \quad (21)$$

The important conclusion that can be drawn from the comparison of (10) and (20) is that both the time constant and the equilibrium particle concentration decrease when the setup is set in motion.

By analogy with (12), the particle deposition velocity for the setup in motion is found from the following expressions:

$$A_{2,i} = \frac{V}{\tau_{2,i}} - Q_0, \quad (22)$$

$$\Delta A_{2,i}^{S_2} = V\frac{\tau_{1,i} - \tau_{2,i}}{\tau_{1,i}\tau_{2,i}} \quad (23)$$

The uncertainties of $A_{2,i}$ and $\Delta A_{2,i}^{S_2}$ are given from:

$$u(A_{2,i}) = \sqrt{\frac{\Delta V^2}{6\tau_{2,i}^2} + \frac{\Delta Q_0^2}{6} + \frac{V^2}{\tau_{2,i}^4}u^2(\tau_{2,i})}, \quad (24)$$

$$u\left(\Delta A_{2,i}^{S_2}\right) = \sqrt{\frac{\Delta V^2}{6\tau_{2,i}^2} + \frac{V^2}{\tau_{2,i}^4}u^2(\tau_{2,i}) + \frac{V^2}{\tau_{2,i}^4}u^2(\tau_{2,i})}. \quad (25)$$

*2.3. General Solution for Particle Concentration Dynamics in an Aerosol Chamber without MSPE*

Assuming that both $A_{1,i}$ and $A_{2,i}$ do not change in time, at least on the time scale of a typical experiment, the dynamics of particle concentration is described by the following first-order ordinary differential equation:

$$V\frac{dC_i}{dt} + C_i(t)\left(Q_0 + A_{1,i} + \Delta A_{2,i}^{S_2}[H(t) - H(t-\zeta)]\right) = Q_0 C_{0,i}, \tag{26}$$

where $H$ is the Heaviside step function, the setup is set in motion in the time interval $t \in [0; \zeta]$. Generally, $C_{0,i}$ can vary with time. However, for the sake of simplicity, we assume it constant during one test.

The general solution of (26) in the time interval $t \in [0; \infty)$ is the following:

$$C_i(t) = \frac{C_{0,i}Q_0}{Q_0 + A_{1,i} + \Delta A_{2,i}^{S_2}[H(t) - H(t-\zeta)]} + C_4 \exp\left(-\frac{Q_0 + A_{1,i} + \Delta A_{2,i}^{S_2}[H(t) - H(t-\zeta)]}{V}t\right) \tag{27}$$

and the integration constant

$$C_4 = \left(C_{1,i}^{eq} - C_{2,i}^{eq}\right)\left(1 + exp\left(-\frac{\zeta}{\tau_{2,i}}\right)H(t-\zeta)\right) \\ + \Delta C_i(0)\left(1 - \left(1 - \exp\left(-\frac{\zeta}{\tau_{2,i}}\right)\right)H(t-\zeta)\right), \tag{28}$$

where $\Delta C_i(0)$ is the deviation of the particle concentration from the equilibrium value $C_{1,i}^{eq}$ at $t = 0$.

If the particle deposition velocity for certain types of aerosols, e.g., artificial aerosols, should be determined experimentally, a method of instant pulse can be employed. It consists in rapid introduction of a certain volume of aerosol into the aerosol chamber. The pulse duration should normally be much shorter than the time constant of the particle concentration transient. Then, the rate of particle injection can be described by the Dirac delta function:

$$K_{im,i} = G_{im,i}\delta(t), \tag{29}$$

where $G_{im,i}$ is the rate of particle injection. Injection takes place at the time $t = 0$.

The transient decay after the pulse follows (9) or (19) depending on whether the setup is at a standstill or it is set in motion, correspondingly. The integration constants $C_2$ and $C_3$ are found from the following expressions:

$$C_2 = C_3 = \frac{G_{im,i}}{V}. \tag{30}$$

The time constants of the transients can be determined using (10) or (20) and then the particle deposition velocities are obtained from (12) or (22).

*2.4. Setup in Motion with MSPE*

To account for the additional source of particle emission due to mechanical stimulation (abrasion, rubbing, drilling, machining, etc.), the rate function of MSPE, $K_{te,i}(t)$, should be added at the right part of the general equation of particle balance (26). We assume that the source of the particle emission can only be active during the mechanical action. This means that the time interval, when mechanical action occurs $[\varphi_0; \varphi_1]$, is contained within the time interval, when the setup is set in motion $[0; \zeta]$. The resulting general equation:

$$V\frac{dC_i}{dt} + C_i(t)\left(Q_0 + A_{1,i} + \Delta A_{2,i}^{S_2}[H(t) - H(t-\zeta)]\right) \\ = Q_0 C_{0,i} + K_{te,i}(t)[H(t-\varphi_0) - H(t-\varphi_1)] \tag{31}$$

has an analytical solution in form (27) in the time interval $[0; \varphi_0]$. However, analytical solution of (31) in the time interval $[\varphi_0; \infty)$ can be found analytically only if $K_{te,i}(t)$ is a known integrable function. In this case, the general solution of (31) is the following:

$$C_i(t) - C_{2,i}^{eq} = C_4 \exp\left(-\frac{t}{\tau_{2,i}}\right), \ t \in [0; \varphi_0) \tag{32}$$

$$C_i(t) - C_{2,i}^{eq} = \left[C_5 + \frac{1}{V} \int K_{te,i}(t - \varphi_0) \exp\left(\frac{t - \varphi_0}{\tau_{2,i}}\right) dt\right] \exp\left(-\frac{t - \varphi_0}{\tau_{2,i}}\right), \ t \in [\varphi_0; \varphi_1) \tag{33}$$

$$C_i(t) - C_{2,i}^{eq} = C_6 \exp\left(-\frac{t - \varphi_1}{\tau_{2,i}}\right), \ t \in [\varphi_1; \zeta) \tag{34}$$

$$C_i(t) - C_{1,i}^{eq} = C_7 \exp\left(-\frac{t - \zeta}{\tau_{1,i}}\right), \ t \in [\zeta; \infty). \tag{35}$$

where $C_4$, $C_5$, $C_6$ and $C_7$ are the appropriate integration constants, which can be found from the initial conditions and taking into account continuity of $C_i$ on the interval boundaries.

In practice, the function $K_{te,i}$ is usually unknown and sought in experiment. The rate of MSPE, $K_{te,I}$, can be found directly from the measured time series of particle concentration within the time interval of mechanical action $[\varphi_0; \varphi_1]$ using the following expression:

$$K_{te,i}(t) = V\frac{dC_i}{dt} + C_i(t)(Q_0 + A_{2,i}) - Q_0 C_{0,i}. \tag{36}$$

## 3. Experimental Setup

An original miniature pin-on disk tribometer equipped with an aerosol chamber was used in this study. The volume of the aerosol chamber, *V*, was 15 l ($u$ ($V$) = 0.28 L). The internal surfaces of the aerosol chamber made of poly (methyl methacrylate) were lined with conductive electrostatic discharge film. A rotatable table was situated in the centre of the chamber (Figure 3). A piece of solid material with the parallelepiped shape and dimensions $50 \times 50 \times 40$ mm$^3$ was attached to the table. It was wrapped into electrostatic discharge film and served to simulate a test piece. The counter body made of carbon steel, which was normally used for rubbing, was separated from the simulated sample during these tests, i.e., no rubbing took place. The rotation velocity could be adjusted between 60 rpm and 600 rpm using a servo drive. A scanning mobility particle sizer (SMPS) was connected to the chamber through a stainless steel pipe. The distance between the inlet of the pipe and the axis of rotatable table could be adjusted. Usually this distance was set 70 mm. The air flow, $Q_0$, at the inlet of SMPS was 0.750 L/min. In order to measure particle deposition velocity with higher accuracy, in several tests the ambient particle concentration was intentionally increased using an electric fan.

Artificial aerosols were generated using three different methods which have been described in detail in literature: (i) nebulization of tap water [28], (ii) cellulose burning [29] and (iii) aerosolization of nano-TiO$_2$ P25 powder [30]. The artificial aerosols were used to measure the particle deposition velocity via an instant pulse method. The important advantage of the proposed method is that it relies on measuring time constant of concentration decay, while the knowledge of aerosol injection rate is not required. The aerosols were injected into the aerosol chamber, approximately 8 cm above the centre of the rotatable table. The duration of the injection was 20 to 30 s, that is considerably shorter than any time constants of observed transient processes.

In the end, the model was employed to quantify the fine particle emission at friction modifier –steel sliding interface using the device shown in Figure 3. A parallelepiped-shape sample of friction modifier was attached to the pin holder and pressed against a steel disk with a normal load of 3.2 N. The projected contact area was around 90 mm$^2$. The friction modifier contained 20 wt.% of MoS$_2$ powder and 1 wt.% multiwall carbon nanotubes (MWCNTs) dispersed in vinylester polymer matrix. Linear sliding velocity was 3.3 m/s and rotation velocity was 1800 rpm.

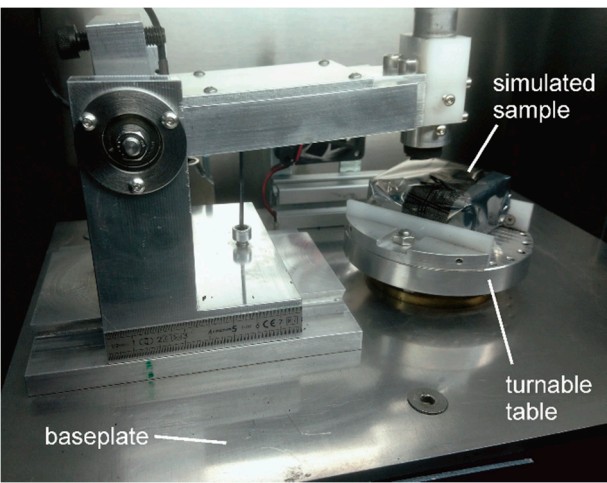

**Figure 3.** A photograph of the miniature pin-on-disk tribometer with the hood of the aerosol chamber removed.

## 4. Results and Discussion

### 4.1. Experiments with Ambient Particle Aerosols

The effect of the table rotation on the concentration of particles of various sizes (in nm) is shown in Figure 4. With the start of rotation, the initially stable particle concentrations, $C_{1,i}^{eq}$, underwent transient decrease towards the lower equilibrium value, $C_{2,i}^{eq}$. The decrease was pronounced in the range 11.5–86.6 nm, but weak in the range 115.5–205.4 nm. Rotation did not cause any notable effect on the concentrations of the particles larger than 205 nm. The time constant of the transient generally increased with increasing particle size. When rotation ended, the particle concentrations returned to their corresponding initial equilibrium values following the second transient. With the setup at a standstill, the time constants of the transient were larger than with the setup in motion.

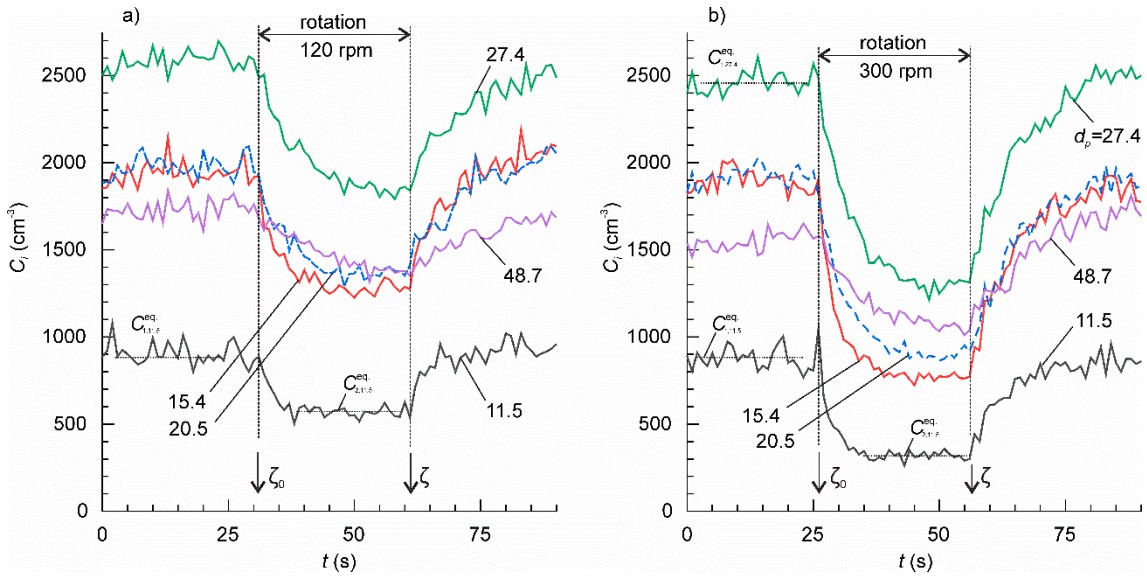

**Figure 4.** The concentration time series for the ambient particles of various sizes. The mean particle size for each channel is shown at the plots (nm). The rotation speed of the table was: (**a**) 120 rpm and (**b**) 300 rpm.

For the initial conditions corresponding to the above experiment:

$$\left[\begin{array}{l} C_i = C_{1,i}^{eq},\ \Delta A_2 = 0,\ t < \zeta_0 \\ \qquad \Delta A_2 \neq 0,\ \zeta_0 \leq t < \zeta \\ \qquad \Delta A_2 = 0,\ t \geq \zeta \end{array}\right., \tag{37}$$

the solution of (1) is a combination of Equations (9) and (19):

$$\left[\begin{array}{l} C_i = C_{2,i}^{eq} + \left(C_{1,i}^{eq} - C_{2,i}^{eq}\right)\exp\left(-\frac{t-\zeta_0}{\tau_{2,i}}\right),\ \zeta_0 \leq t < \zeta \\ C_i = C_{1,i}^{eq} - \left(C_{1,i}^{eq} - C_{2,i}^{eq}\right)\left(1 - \exp\left(-\frac{\zeta-\zeta_0}{\tau_{2,i}}\right)\right)\exp\left(-\frac{t-\zeta}{\tau_{1,i}}\right),\ t \geq \zeta \end{array}\right., \tag{38}$$

where $\zeta_0$ and $\zeta$ are times when the table rotation began and ended, correspondingly.

The time constants $\tau_{2,i}$ and $\tau_{1,i}$ of the two transients can be found from the slopes of the linear fitting of experimental time series on a semi-log plot $\ln(C_i - C_i^{eq})$ vs. $t$. The data, which could not be satisfactorily approximated by a linear function, i.e., when the adjusted coefficient of determination $R_{adj}^2 < 0.5$, were discarded from further analysis. The results are shown in Figure 5a,b. As we expected, $\tau_{1,i}$ was nearly constant independently of the rotation velocity for each particle size, $d_p$. This supports our hypothesis about the additive property of the particle deposition rates (18). The mean value of $\tau_{1,i}$ for each $d_p$ was determined by averaging the data in various tests. A linear function excellently fitted mean $\tau_{1,i}$ on a semi-log plot $\tau_{1,i}$—$\log d_p$ yielding $R_{adj}^2 = 0.9846$. The slope of the linear function $\frac{d\tau_1}{d(\log d_p)}$ was $1074 \pm 50.6$. The $\tau_{2,i}$, exhibited similar linear trend in a semi-log plot, but the slope tended to decrease with increasing rotation velocity. Except for a few data points with large dispersion, $\tau_{2,i}$ was smaller than $\tau_{1,i}$.

The $A_{1,i}$ and $A_{2,i}$ were determined using Equations (12) and (22), correspondingly. The results are shown in Figure 5c,d. The datasets were fitted by linear functions on a log-log plot. For all $d_p$ $\Delta A_{2,i}$ was positive and increased as rotation velocity increased. The increase was more significant for larger particles. The particle deposition velocity increased between 52% and 1350% with respect to the setup at a standstill. So far, similar behaviour $A_i \propto d_{p,i}^{-b}$ has been reported in a number of studies [21,25]. It means that particle deposition is controlled mainly by diffusion since the deposition rate decreases as the particle diameter increases [26]. The increase of the deposition velocity with the air flow velocity observed in the present work also agrees well with Lai and Nazaroff [25] model.

The nonlinear increase of the particle deposition velocity with the rotation velocity suggests that there should be a threshold, below which the deposition rate is independent of the rotation velocity. This threshold must rely on the onset of turbulent regime of the air flow. Appelquist et al. [31] determined the critical Reynolds number for the onset of global instability of the rotating disk flow $Re_{cr} = 583$. The transition from transitional to turbulent flow usually occurs for Reynolds numbers between 2300 and 4000. The corresponding critical rotation velocity can be found from the following expression [32,33]:

$$n_{cr} = \frac{30\nu Re_{cr}}{\pi r^2}\ [\text{rpm}], \tag{39}$$

where $\nu$ is the kinematic viscosity of air, $r$ is the radius of the rotatable table. In our case, $r = 0.1$ m and $\nu = 14.6 \times 10^{-6}$ m$^2$/s [34], then $n_{cr|Re_{cr}=583} = 8.1$ rpm and for transitional–turbulence transition $n_{cr|turb}$ is in the range 32– 56 rpm. The latter reasonably agrees with our experiments, in which at 30 rpm any variations of particle concentration could not be distinguished during table rotation.

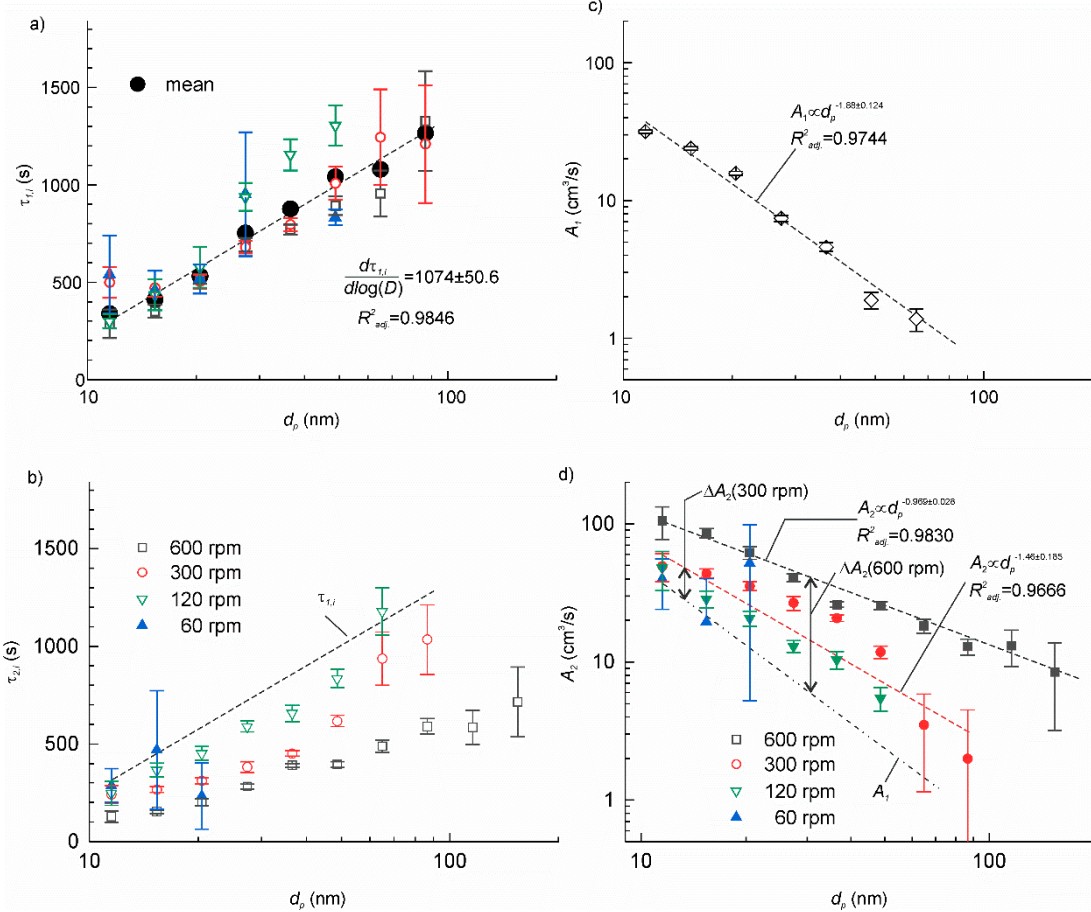

**Figure 5.** (**a**) The time constant of the transient with the setup at a standstill. The small symbols correspond to the experimental data obtained at various rotation velocities. The large solid circles are the mean values at each particle size; (**b**) the time constants of the transient with rotation on as function of the particle mean size and the rotation velocity. The dashed line is the linear approximation of the mean $\tau_{2,i}$ from panel (**a**); (**c**) the particle deposition velocity for the setup at a standstill. The dashed line shows the linear fit of the data on a log-log plot; (**d**) the particle deposition velocity with rotation on. The dashed lines show the linear fit of the data on a log-log plot at 300 and 600 rpm. The dash-dotted line is the linear fit of $A_{1,i}$ from panel (**c**).

## 4.2. Dynamics of Artificial Nanoparticle Aerosols

The time constants of particle deposition in the setup at a standstill for three types of artificial particles and the reference ambient airborne particles are plotted in Figure 6a. It is noteworthy that the particle deposition velocity is specific to the type of particles. All plots showed nearly linear dependence, but the slopes of $d\tau_{1,i}/d(\log d_p)$ varied significantly. For the particles from tap water and cellulose smoke, the slopes were close to zero (slightly positive for tap water and slightly negative for smoke). For aerosols of nano-TiO$_2$ P25, the slope was negative.

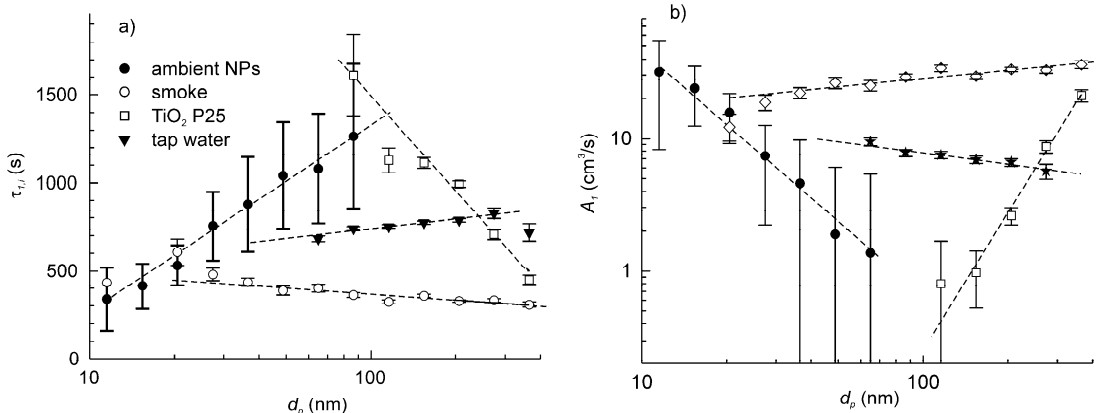

**Figure 6.** The dependence of time constant of particle deposition (**a**) and particle deposition velocity (**b**) on the particle size for three types of artificial particle aerosols and ambient airborne particles. The data correspond to the setup at a standstill.

Although understanding the reasons for the different behaviours of the particles was not among the objectives of this study, some relevant conclusions can be drawn. Various studies reported that the particle deposition velocity is an U-shape [21,23,27] or V-shape [25,35] function of the particle aerodynamic diameter. So, we can suggest that the experimental plots of $A_1(d_p)$ correspond to different parts of the same generalized dependence. In other words, the experimental plots are shifted relative to each other along the axis of abscissa. This could be attributed to the differences in dynamic shape factors between the particles of different types [36]. The dynamic shape factor enters as a scale factor in the expression for transformation of the volumetric diameter to the aerodynamic diameter. On the other hand, the particle specific weight does not seem to be a critical parameter, since three out of four types of particles—the ambient particles, the tap water particles and smoke—should have quite similar density ranged between 2.0 and 2.7 g/cm$^{-3}$, but they showed very different deposition behaviour. The particle densities were estimated considering their main constituents, which include mineral salts, silica, light metal oxides, soot, etc. In turn, the density of TiO$_2$ was about two-fold greater than that of the other species.

The particle deposition velocity for the setup at a standstill was calculated for each type of particles using Equation (12) and is shown in Figure 6b. Then, the measurements were repeated but with the table rotating at 600 rpm. Figure 7a shows the example of the plots of time constants $\tau_{1,i}$ and $\tau_{2,i}$ for the ambient airborne particle and the tap water aerosol, whereas Figure 7b shows the corresponding particle deposition velocities. Like for ambient airborne particles, the artificial aerosol exhibited considerable increase in deposition velocity when the setup was set in motion. This increase, $\Delta A_{2,i}^{S2}$, ranged between 30% and 200% of corresponding $A_{1,i}$ depending on the particle size. The plots of $A_{2,i}$ vs. log $d_p$ had similar behaviour for all particle types.

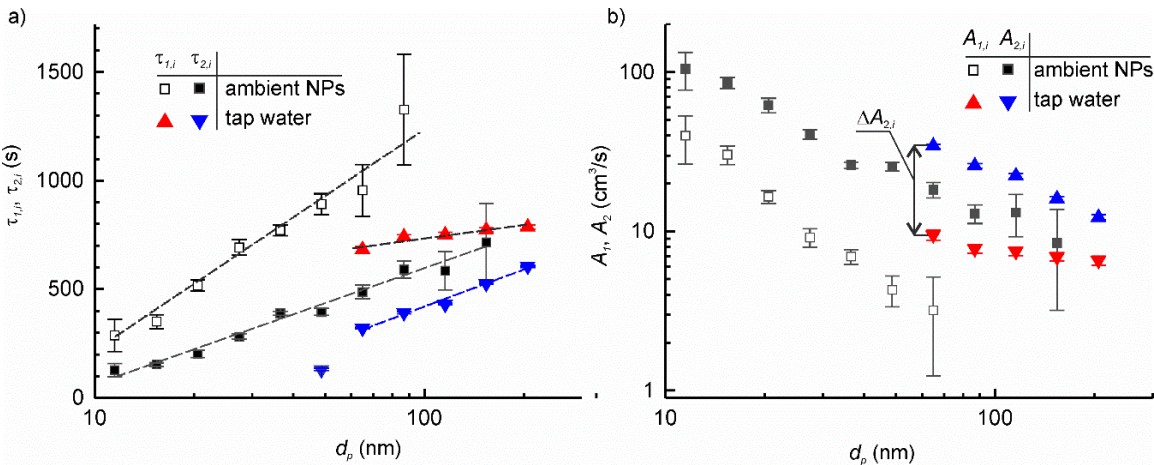

**Figure 7.** The dependence of time constant of particle deposition (**a**) and particle deposition velocity (**b**) on the particle size with the setup at a standstill and in motion for particle aerosols obtained by nebulization of tap water and ambient airborne particles.

### 4.3. Simulation of MSPE by Injection of Aerosols

The following experiment was conducted to demonstrate how the developed model can be employed in the analysis of time series of intrinsically unsteady MSPE. The experiment involved five phases: standstill, table rotation, table rotation with aerosol injection, table rotation, and standstill—as shown in Figure 8. An artificial aerosol was prepared from the tap water as described before. The aerosol was injected in two pulses, three minutes each, of different rates with a one-minute pause between them. The measured time series for six channels of the particle sizer are shown in Figure 8a.

The rate of particle injection (Figure 8b) was calculated from (37) using experimentally determined values of $A_{2,i}$. The injection pulses are clearly resolved for all channels. The correctness of the values of $\tau_1$ or $\tau_2$ used for calculation is supported by the observation that the calculated emission rate returns to zero both between the pulses and in the end of the test. In case of incorrect values of these parameters, the "zero" rate would be positive or negative that would be inconsistent with the experiment.

The important consideration in the analysis of the MSPE is that the instant particle emission rate is not the material characteristics solely but depends also on the configuration of the experimental setup and the operational parameters. Furthermore, the particle emission rate in real experiments showed notable scattering caused by the intrinsically dynamic nature of MSPE characterized by sharp bursts and transitions [37]. Differentiation of noisy experimental data can further increase scattering of results. A different parameter—a cumulative particles emission, $N_{te,i}(t)$—can be a good alternative for characterization of MSPE. It can be determined by integration of the instant MSPE rate:

$$N_{te,i}(t) = \int_{\varphi_0}^{t} K_{te,i}(t)\mathrm{d}t = V\left(C_i(t) - C_i(\varphi_0) + \frac{1}{\tau_{2,i}} \int_{\varphi_0}^{t} \left(C_i - C_{2,i}^{eq}\right)dt\right),\ \varphi_0 < t \le \varphi_1. \tag{40}$$

The plots of cumulative number of emitted particles are shown in Figure 8c. The contribution from each emission pulse is clear. The value at time $\varphi_1$ is the total number of the particles emitted in the experiment. The coefficient of variation for $N_{te,i}$ is much smaller than for the maximum emission rate. To obtain the material specific characteristics of MSPE, $N_{te,i}$ can be further normalized by the duration of the test, applied load, the surface area from which emission took place, etc.

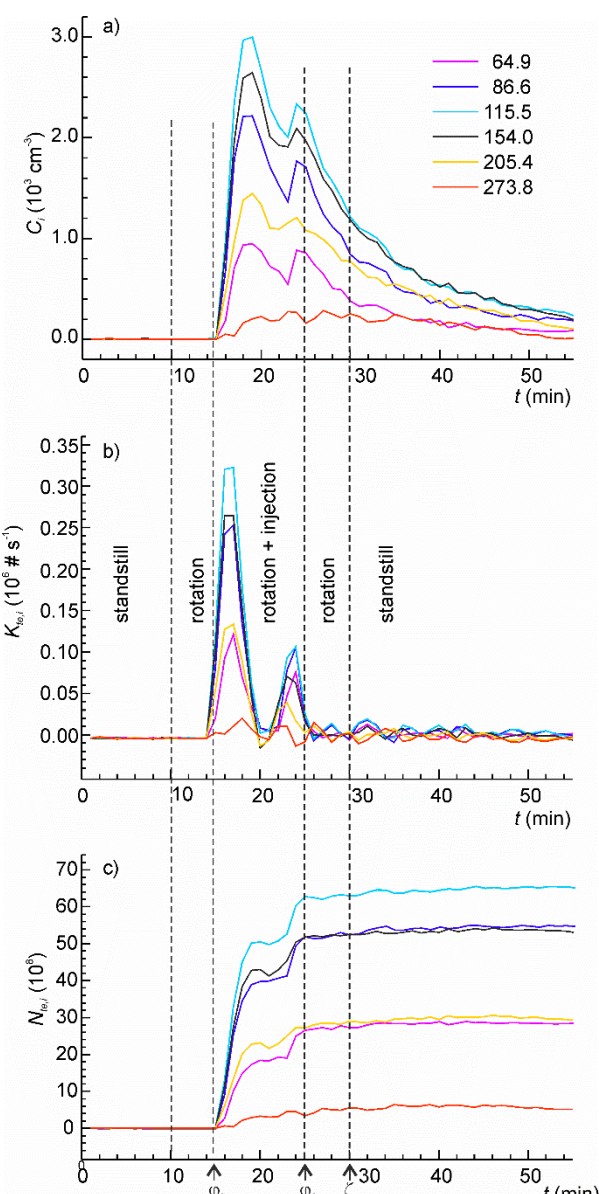

**Figure 8.** (**a**) The time series of particle concentrations in the aerosol chamber in the mechanically stimulated particle emission (MSPE) simulation test using artificial aerosol obtained using nebulization of tap water; (**b**) the instantaneous rate of particle injection determined from the concentration time series; (**c**) the total cumulative number of injected particles as function of time.

### 4.4. Objective Quantification of Fine Particles Triboemission at Friction Modifier–Steel Sliding Contact

The time series of fine particle concentrations registered in the experiment are shown in Figure 9a. After filtering the ambient air, the concentration of residual aerosols was very low, around 3 cm$^{-3}$, with the mode centred at 27 nm. As soon as rotation started, the concentration gradually increased exceeding 10 cm$^{-3}$, whereas the mode shifted to 115–154 nm range. At the same time, the concentration of ultrafine particles decreased (Figure 9b,c). After the rubbing stopped, the concentrations gradually returned to the initial background value. The time constants of the transitional decays were determined by fitting the time series using exponential functions. Then, the particle deposition velocities at standstill were evaluated following the method described in Section 2.1. The results are summarized in Table 1. For the sake of simplicity and given the experimental imitations for determining actual values of $A_{2,j}$, we adopted that with the setup in motion the particle deposition velocity in this test

increased the same factor as in our previous test with tap water aerosols. This assumption is backed up by the similarity of degree of accelerated deposition of ultrafine particles in both experiments.

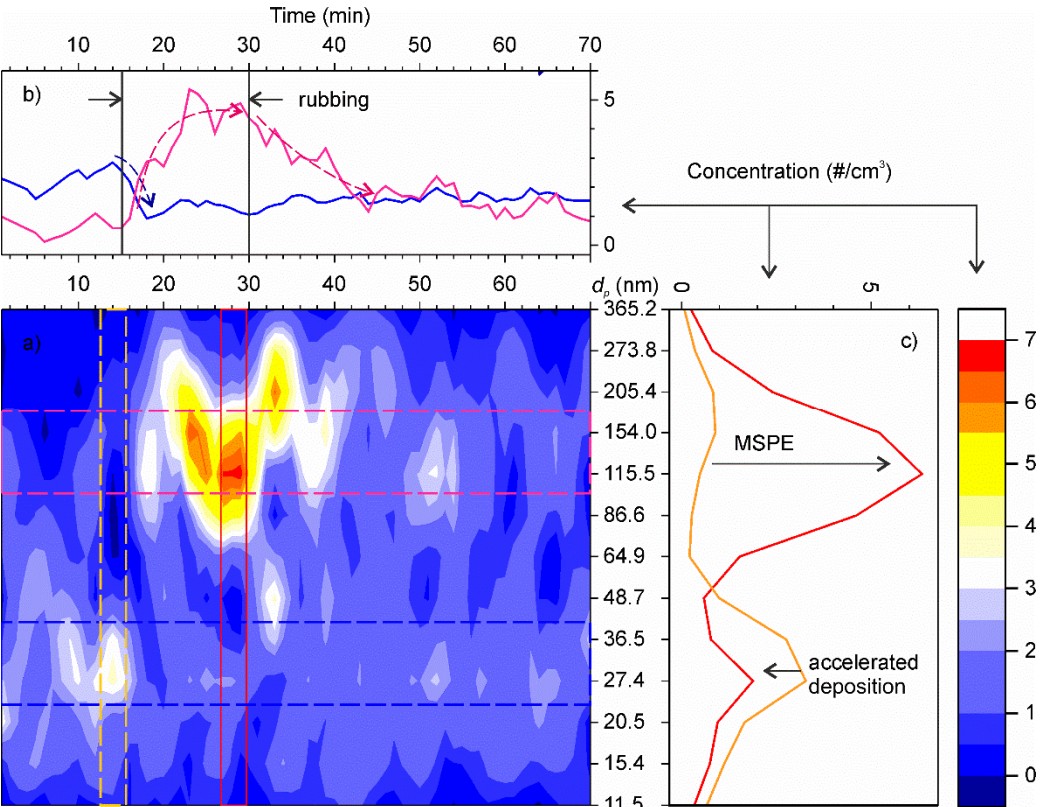

**Figure 9.** (**a**) Time series of particle number concentrations as function of particle size. The colour scale shows the number concentrations. (**b**) Average time series profiles centred at 27 nm (blue line) and 130 nm (pink line). Dashed lines show the main trends. (**c**) Distribution of aerosol particles by size before the rubbing start (orange) and during rubbing (red).

**Table 1.** Results of objective quantification of fine particle emission at sliding contact of friction modifier against steel.

| $d_p$ (nm) | $\tau_{1,j}$ (s) | $A_{1,j}$ (cm³/s) | $A_{2,j}$ (cm³/s) | $\overline{K_{te,i}}$ (#/s) |
|---|---|---|---|---|
| 86.6 | 1137 ± 323 | 0.693 ± 0.197 | 2.30 ± 0.653 | 29.8 ± 6.90 |
| 115.5 | 1185 ± 160 | 0.158 ± 0.021 | 0.470 ± 0.0638 | 41.7 ± 8.80 |
| 154.0 | 1055 ± 107 | 1.72 ± 0.177 | 3.98 ± 0.410 | 50.6 ± 9.11 |
| 205.4 | 797 ± 205 | 6.32 ± 1.63 | 11.8 ± 3.04 | 60.4 ± 10.8 |
| 273.8 | 1123 ± 232 | 0.860 ± 0.178 | - | - |

The rate of particle emission was assessed using a simplified expression (40): the instant concentration was replaced by its average during rubbing. Thus, the derivative of mean concentration is null:

$$\overline{K_{te,i}} = \overline{C_i}(Q_0 + A_{2,i}) - Q_0 C_{0,i}. \tag{41}$$

*4.5. A Groundwork for the Development of the Common Standard Method and Procedure for Objective Quantification of Kinetic Parameters of Fine Particles Triboemission*

The findings of this work have shown that ignoring particle deposition during the experiment can result in significant underestimation of the emission rate. In this study, the "hidden" portion of the emission rate was 3.6% to 49% of the apparent one, which is the product of the mean number particle concentration, $\overline{C_i}$, and the air flux, $Q_0$. These figures can be even higher for the rotating tables of

larger size since the critical linear velocity, at which turbulisation begins, is inversely proportional to the table radius (see (38)). It should be stressed that the relevant parameter for the critical velocity is not the diameter of the friction zone, but the diameter of the rotatable table itself. This raises a number of questions as to the design of the experimental setups, choice of the experimental conditions, data processing and reporting the results. Although significant further efforts are needed to establish standard methods for accurate and objective measuring of aerosol particles triboemission, the developed model provides a necessary blueprint for a round robin test, which is summarized as follows:

A.   Experimental setup

- The experimental setup must have an aerosol-tight hood impeding uncontrolled dispersion of the emitted particles in and income of the particles from the environment.
- The concentration of aerosol particle in the surrounding atmosphere should be measured or controlled.
- The rotation speed and the dimensions of the moving parts must be specified.

B.   Experimental procedure

- A preliminary test aimed at determining the deposition velocities $A_{1,I}$ and $A_{2,I}$ of relevant aerosols must be conducted (Figure 10).
- Time series of aerosol particle concentrations must be measured before, during and after mechanical solicitation in order to determine the background particle concentrations and to measure the kinetic parameters of concentration decays.

C.   Data processing and reporting the results

- The rates of aerosol particles triboemission must be calculated considering the deposition velocity for each specific experimental setup and experimental conditions (rotation or linear velocity, speed profile, etc.).
- The report should include description of the geometry and dimensions of the moving parts of the setup, their rotation and linear speed, and (optionally) the deposition velocities for the relevant aerosol particles for the experimental setup.

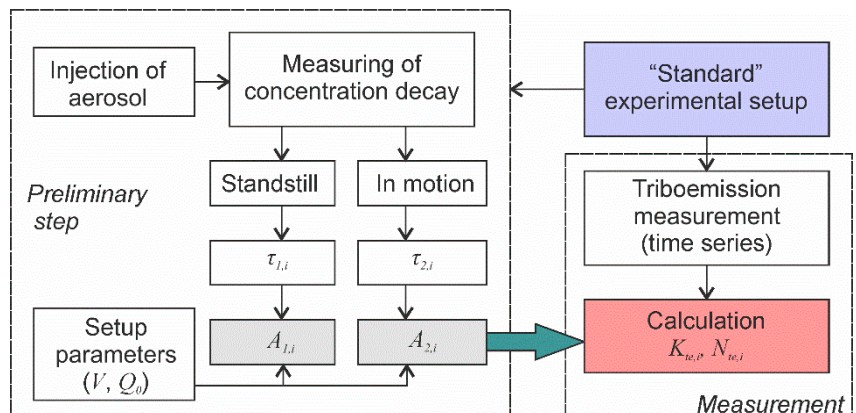

**Figure 10.** The algorithm of the experimental procedure for objective and accurate quantification of particle triboemission.

In the future, we hope the forces of various research groups can be combined to verify the methodology put forward in this work on various setup configurations and for different aerosols as well as to pursue transition to a mechanistic description of aerosol deposition processes in the experimental tribological setups.

## 5. Conclusions

Our experiments demonstrated that air turbulization by moving parts of the experimental setup can significantly affect the measured nanoparticle emission rate resulting in underestimated results. A model of the aerosol particle dynamics in the aerosol chamber was developed for accurate and objective quantification of the particle triboemission rate. The model is based on the equation of particle balance in the chamber. The influence of various setup and process parameters was analyzed.

It was found that the dynamics of aerosol particle concentration obeys first order exponential approach with two time constants corresponding to the setup at a standstill and the setup in motion, respectively. The latter one was significantly smaller than the former one because of the enhanced particle deposition caused by moving components. The model introduced setup-specific empirical parameters characterizing the particle deposition velocity, which can be found from a simple relationship involving experimental parameters and the time constant of particle concentration transients. The particle deposition velocity is a function of the particle size, velocity of the moving elements and the type of the aerosol particles. Once the particle deposition velocities are evaluated, the developed model can be applied for the analysis of the time series of mechanically stimulated particle emission and calculation of real instantaneous emission rates or the cumulative number of emitted particles. The suitability of the model for this purpose was demonstrated in a series of simulated experiments.

The developed model for evaluation of the real particle emission rate paves the way to define a standard method for the quantitative characterization of mechanically stimulated particle emission in various materials, enabling studies to be compared.

**Author Contributions:** Conceptualization, R.N.; Methodology, R.N. and M.C.; Validation, J.A.C.C. and R.N.; Formal Analysis, R.N. and A.T.; Writing—Original Draft Preparation, R.N.; Writing—Review & Editing, M.C., J.A.C.C., A.T.; Supervision, A.T.; Funding Acquisition and Project Administration, M.C., R.N. and A.T. All authors have read and agreed to the published version of the manuscript.

**Funding:** This research was funded by European Union's Horizon 2020 research and innovation programme under the Grant Agreement No LIFE13 ENV/ES/001221 and by the Spanish Ministry of Science, Innovation and Universities through the grants BIA2016-79582-R and EIN2019-102889 and Spanish National Research Council through the grant COOPB20363.

**Acknowledgments:** The authors acknowledge F.R. for his assistance in development of the experimental setup and J.S. and R.B. from the Group of Advanced Materials and Energy of Metropolitan Technological Institute of Medellin for providing the samples of solid lubricants.

**Conflicts of Interest:** The authors declare no conflict of interest.

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
