# Peer review of "Triboemission of FINE and Ultrafine Aerosol Particles: A New Approach for Measurement and Accurate Quantification"

_lubricants, doi:10.3390/lubricants8020021_

Round 1
Reviewer 1 Report
This is the comments on the paper N ID: lubricants-715426 submitted to “Lubricants”.
Title: Triboemission of fine and ultrafine aerosol particles: a new approach
for measurement and accurate quantification.
Journal: Lubricants.
This manuscript has been quickly reviewed by reviewer and the comments are attached at the bottom.
Rate the Manuscript:
Significance to field and specialization of “Lubricants” journal: good.It has been experimentally established that air turbulisation by moving parts of the experimental setup can significantly affect the measured nanoparticle emission rate resulting in underestimated results. A model of the aerosol particle dynamics in the aerosol chamber was developed for accurate and objective quantification of the particle triboemission rate. The model is based on the equation of particle balance in the chamber. The influence of various setup and process parameters was analysed. It was found that the dynamics of solid aerosol particle concentration obeys first order exponential approach with two time constants corresponding to the setup at a standstill and the setup in motion, respectively. The latter one was significantly smaller than the former one because of the enhanced particle deposition caused by moving components. The model introduced setup-specific empirical parameters characterising the particle deposition velocity, which can be found from a simple relationship involving experimental parameters and the time constant of particle concentration transients. The particle deposition velocity is a function of the particle size, velocity of the moving elements and the type of the aerosol particles. The applicability of the model for this purpose was demonstrated in a series of simulated experiments.
Originality: good.
Clarity and presentation: acceptable. Appropriateness for Journal: appropriate subject mater for the journal “Lubricants”. Need for rapid publication: yes. Recommendations: to sent after minor revision to “Lubricants”.
However, the following remarks and questions were arisen after reading of the manuscript:
Abstract should include only main statements and conclusions of study. Please, re-write this section in more compact and clear view. Section “CONCLUSIONS”: Text should be improved. All abbreviations should be explained. Please provide the appropriate references for some equations, because the readers may indentify them as ‘written in first time’.The literature review is weak with limited references, which have experimental results for comparison of theoretical model. May be consider the next new papers:
Wear resistance of hydrogenated high nitrogen steel at dry and solid state lubricants assistant friction, August 2019, Archives of Materials Science and Engineering 2(98):57-67. DOI: 10.5604/01.3001.0013.4607 https://archivesmse.org/resources/html/article/details?id=193096, Influence of the thermal resistance of liquid coolants on the machining of 12Kh18АG18SH steel. Materials Science. 2016. Vol. 52, № 2. P. 200–208. DOI 10.1007/s11003-016-9944-y Final conclusion. All components of the manuscript should be carefully checked: text, figures, equations and tables. Revised version of paper should be prepared and submitted again. 6.2.2020.Author Response
We acknowledge the comments and suggestions proposed by the reviewer to improve the manuscript. The Abstract and Conclusions sections were amended to improve the clarity. The abbreviation in the Conclusion sections was explained. The literature survey was apdated.
Reviewer 2 Report
This paper deals with mechanically stimulated fine aerosol particles. It particularly focusses on how to reach a quantitative measurement of their emission rates in a test chamber. Based on observations that particles deposition velocity has to be considered in order obtain accurate data, the authors developed a dynamic model to take this process into account. The model is based on mass balance and reports the effect of air tubulisation near the chamber walls on the deposition velocity. Additionally, an experimental study has been carried on a dedicated test rig. It provides experimental data to the model in order to physically interpret results.
In my opinion, this is a very interesting paper. Despite a widespread use of engineered nanomaterials, there is not yet a consensus on a methodology to evaluate the impact of fine particles on environment. This paper provides an innovative protocol that could be used for various geometries and materials. This paves the way to define a standard test enabling studies to be compared.
The report is very well written and both experimental protocol and model are clearly presented. One particularly appreciates the authors’ efforts to introduce each step of modelling process and honestly indicate benefits and limitations of the approach. Bibliography is exhaustive and includes the most relevant papers.
For all these reasons I suggest this paper should published as it is.
Author Response
We highly appreciate the reviewer's comments.